# An Investigation of the Influence on Compacted Snow Hardness by Density, Temperature and Punch Head Velocity

Qiuming Zhao [1], Zhijun Li [1,*], Peng Lu [1], Qingkai Wang [1], Jie Wei [1], Shengbo Hu [1] and Haorong Yang [2]

1 State Key Laboratory of Coastal and Offshore Engineering, Dalian University of Technology, Dalian 116024, China; qiumingzhao@mail.dlut.edu.cn (Q.Z.); lupeng@dlut.edu.cn (P.L.); wangqingkai@dlut.edu.cn (Q.W.); jiewei@mail.dlut.edu.cn (J.W.); hushengboandy@163.com (S.H.)
2 China Road and Bridge Corporation, Beijing 100011, China; yanghr1997@163.com
* Correspondence: lizhijun@dlut.edu.cn

**Abstract:** The density, temperature, and punch head velocity are dominant factors to the variation of the compacted snow hardness measured by penetrometers. This effect is essential to the construction and operation of compacted snow roads. The Improved Motor-driven Snow Penetrometers (IMSP) are utilized in this research to control the penetration speed and measure the true cone hardness during snow penetration. This study employs a multi-method approach combining orthogonal experiments and the Support Vector Regression (SVR) technique to analyze the effects of these three factors on snow hardness. The results of this investigation indicate that, under identical conditions, density is positively correlated with the hardness of compacted snow, and its sensitivity and significance to the compacted snow hardness are the greatest. Temperature and penetration speed have an effect on hardness, which cannot be completely ignored. The hardness of snow close to its melting point, regardless of its density, decreases significantly at high penetration rates. This study investigates the factors that influence the hardness of compacted snow and provides substantial technical support for the design, construction, and maintenance of snow roads.

**Keywords:** ice and snow structures; compacted snow hardness factors; compacted snow density; orthogonal experiment; the Support Vector Regression (SVR)





## 1. Introduction

Recent trends in ice and snow sports have led to a proliferation of studies that focusing on the mechanical properties of compacted snow. As an important mechanical property of compacted snow, the hardness plays a significant role in determining the skiing road performance, because the penetration ability, friction force, and apparent contact area of skis are closely related to the snow hardness [1–3].

Up to now, great efforts have been devoted to exploring the variation of snow mechanical property, with a focus on the influence by temperature, loading rate, density, and other parameters. Gold [4] utilized the Snow Hardness Gauge, National Research Council type, to investigate the snow density, temperature, and other factors that may affect the hardness of snow. Subsequently, the relationship between snow hardness and density, as well as the relationship between snow hardness and temperature, were proposed. Schweizer et al. [5] found most mechanical characteristics of snow were shown to be rate-dependent. Temperature had a significant impact on toughness, failure strain, and the amount of deformation required before failure starts. Landauer [6] conducted snow uniaxial compression experiments using two distinct methods: constant strain rate and constant load, and believed that the strain rate of snow was related to snow density, force, and temperature. Lintzén et al. [7] performed a study on the mechanical properties of machine-made snow and found that old machine-made snow has lower strength than new machine-made snow. In general, the uniaxial strength of machine-made snow with the same density as natural snow is comparable. Surveys such as that conducted by

Wang et al. [8] have shown that as the loading rate increased, the uniaxial compressive strength of compacted snow first increased and then decreased. With increasing density, the uniaxial compressive strength of compacted snow increases as well. In addition, as the temperature decreases, snow's uniaxial compressive strength increases, particularly under low loading rates and high density. The uniaxial compressive strength of snow cover is significantly influenced by loading rate, density, and temperature.

The existing body of research on compacted snow hardness suggests that the density of compacted snow, the temperature, and the velocity of the compaction punch head cone are dominant factors. However, little attention has been paid to which one of these three elements is most sensitive and significant. A systematic understanding of how these three factors simultaneously affect compacted snow hardness is still lacking.

In order to contribute to the solution of the problems described above, this study employees the orthogonal experiments and the Support Vector Regression (SVR with orthogonal test data) focused on these three factors: compacted snow density, temperature, and punch cone penetration speed. The orthogonal test method is selected for its validity and efficiency [9,10]. The SVR technique has a number of attractive features: SVR can efficiently manage a nonlinear regression problem by projecting the original feature into a kernel space where data can be linearly discriminated [11,12]. A further advantage of SVR is that it learns a model to characterize the importance of a variable between input and output, whereas a traditional regression method requires the assumption of an inaccurate model [13]. It is an efficient method to incorporate orthogonal experiments into the analysis of SVR, which makes the model high-accuracy without being overly complex [14]. This study investigated the combined impact of these three factors on snow hardness and provided valuable information for the construction and application of compacted snow structures.

## 2. Materials and Methods

### 2.1. Overview of the Research Area

This experiment was conducted from mid-December 2022 to late January 2023 in Bin County, Harbin. Bin County, which is located in the Northeast China Plain, is a typical example of the temperate monsoon climate. Winter lasts for a long time, with an average monthly temperature of −15.8 °C. The snowfall season runs from November to January. The maximal snow depth is approximately 41 cm, and the average annual snowfall is 23.6 mm [8]. In order to avoid human influence as much as possible, the snow material is obtained from the experiment site (45°59′ N, 127°15′ E) far away from the urban district (as shown in Figure 1).

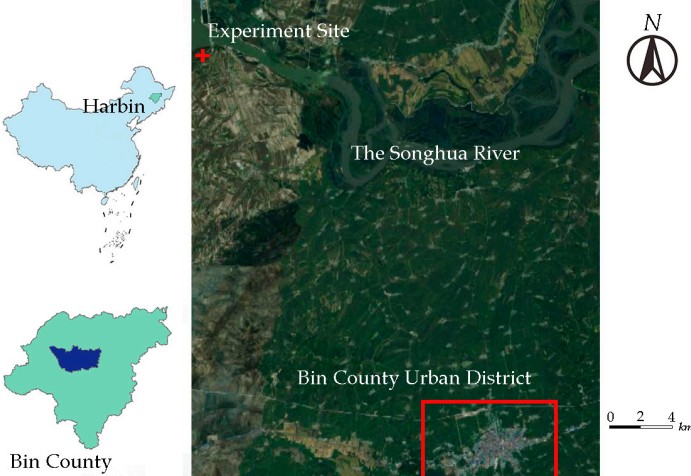

**Figure 1.** Schematic diagram of experimental geographical location. The red box is Bin County urban district.

## 2.2. Research Equipment

The hardness value of snow is measured by the Improved Motor-driven Snow Penetrometer (IMSP) [15]. With traditional mechanical penetrometers (such as the Rammsonde), it is hard to avoid systematical errors caused by different operators. This part of systematical error will be avoided by the IMSP.

Figure 2 illustrates the components of the IMSP instrument.

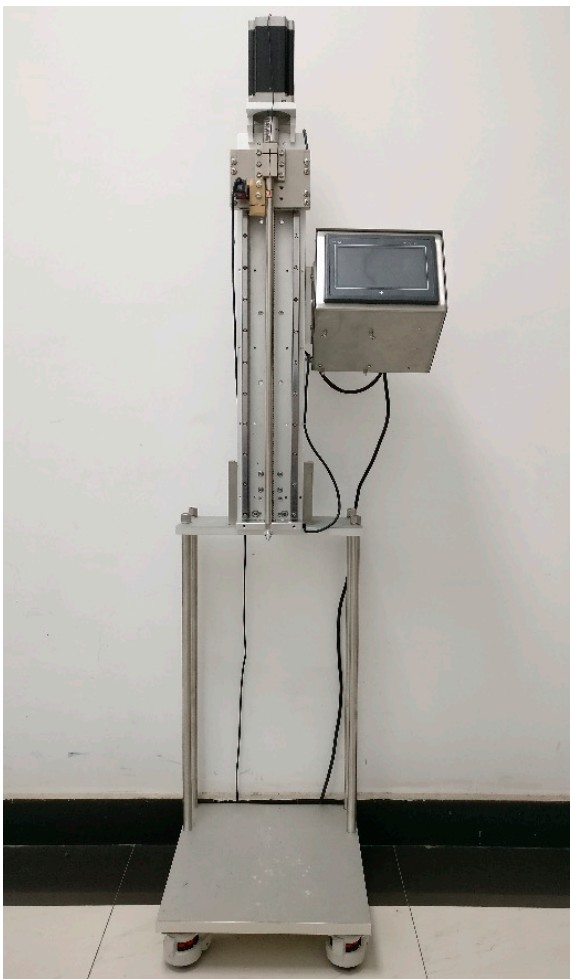

**Figure 2.** Improved Motor-driven Snow Penetrometer.

The IMSP device is composed of top and bottom parts. The bottom part is a bearing table attached to the column that supports the snow sample. A stepper motor, a penetrating rod, and a control box make up the top part. The laser displacement sensor can be sliding downward with the penetrating rod. A pressure sensor is attached to the top end of the penetrating rod, and the cone is a 60-degree punch head.

During the usage period, the IMSP is connected to a 220 V AC power source, and the penetration pace is input by the control box. Driven by a stepper motor, the rod penetrates the snow sample downward at a uniform speed. When the cone comes into contact with the surface of the snow sample, the laser displacement sensor and pressure sensor all begin to effectively collect data. A digital signal which is received by the computer is obtained after the conversion of the response of the sensor. An Ethernet data-acquisition program was available to synchronously store the data of force and displacement, and obtain the relationship between the hardness true value and displacement [15].

## 3. Orthogonal Test Method

### 3.1. Factors Affecting the Hardness of Compacted Snow

Snow metamorphism is usually carried out in stages. At the first stage of metamorphism, the hardness of the compacted snow was found to increase rapidly with increasing growth of bonds in snow. After the first stage of metamorphism, the growth rate of bonds decreases so that snow hardness grows slowly [16]. The particle size of the experimental snow crystals is primarily around 0.6 mm. On a mesoscopic scale, the majority of snow crystals are broken, and the growth of bonds has attained a certain magnitude (as depicted in Figure 3). At this stage, the snow crystal has undergone metamorphosis to some extent, and this state is significantly more stable than that of fresh snow. Therefore, this study indicates that variations in snow crystals during the experiment will have little effect on the snow hardness.

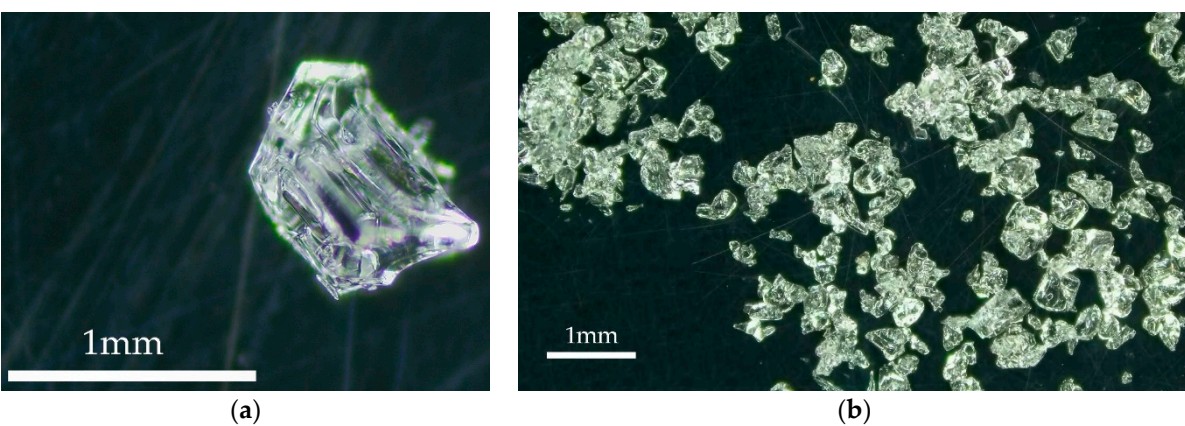

(a) (b)

**Figure 3.** Snow crystal after the first stage of metamorphism: (**a**) Single snow crystal; (**b**) Multiple snow crystals.

Collect snowfall from the experimental site that has undergone the first stage of metamorphism, and use the mass/volume method to control the density of different snow samples. The operation of the mass/volume method is shown in Figure 4a. The test boxes used in this experiment are wooden and detachable. The inner wall section of the test box is 0.25 m × 0.25 m. Layering compaction can guarantee that the total snow sample is as uniform as possible [17]. The compacted sample preparation is shown in Figure 4b.

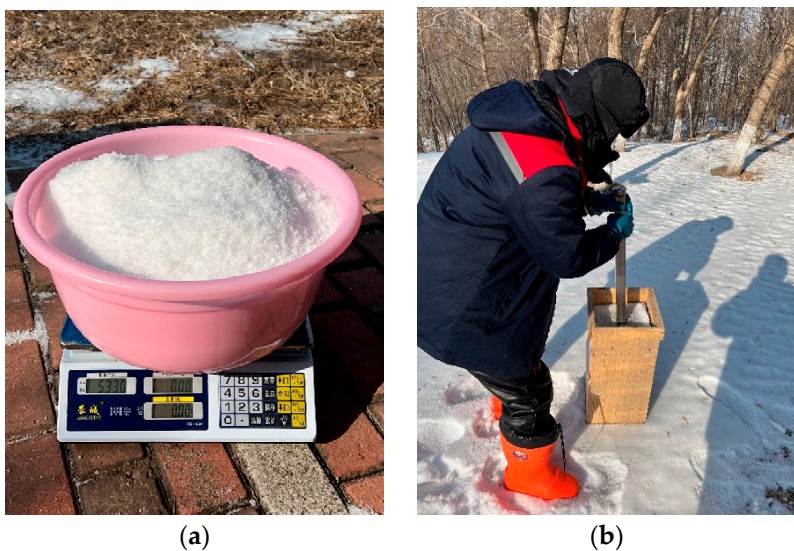

(a) (b)

**Figure 4.** Compacted snow sample: (**a**) Control the density of the snow sample with a high-precision electronic scale; (**b**) Preparation of snow samples by layering compaction.

With a temperature-controlling refrigerator, the prepared snow samples are stored. The IMSP control box manages the cone penetration speed.

### 3.2. Arrangement and Results of Orthogonal Experiment

Applying the orthogonal experimental technique, three-factor, eight-level test was carried out, and the input factors were optionally corresponding to the three columns of the $L_{64}(8^9)$ orthogonal table. The constant penetration velocity of the IMSP punch head, the density, and the temperature of the compacted snow sample are the three factors. The minimum constant penetration velocity of the IMSP punch head is 15.03 mm/s and the maximum is 21.11 mm/s; the minimum density of compacted snow samples is 354 kg/m$^3$ and the maximum is 438 kg/m$^3$. The minimum temperature value is $-38$ °C, and the maximum is $-3$ °C. With equal spacing between each level, these three factors are fixed at eight levels. $V$ represents the cone penetration speed; $\rho$ represents the density of compacted snow samples; and $T$ represents the temperature of compacted snow samples. The design method and experimental factors, in order, are detailed in Table 1.

**Table 1.** Level number of factors for compacted snow hardness test.

| Level Number | Factors | | |
| --- | --- | --- | --- |
| | $V$ (mm/s) | $\rho$ (kg/m$^3$) | $T$ (°C) |
| 1 | 15.03 | 426 | $-3$ |
| 2 | 15.90 | 438 | $-8$ |
| 3 | 16.77 | 354 | $-13$ |
| 4 | 17.64 | 366 | $-18$ |
| 5 | 18.50 | 378 | $-23$ |
| 6 | 19.37 | 390 | $-28$ |
| 7 | 20.39 | 402 | $-33$ |
| 8 | 21.11 | 414 | $-38$ |

Table 2 details the specific experimental design, including experimental factor settings and experimental outcomes (the IMSP Value). Among them, $e$ represents the empty column of the orthogonal experimental table.

**Table 2.** Orthogonal experiment arrangement and test results.

| Test Number | Factor Level | | | Empty Column | IMSP Value (kPa) | Test Number | Factor Level | | | Empty Column | IMSP Value (kPa) |
| --- | --- | --- | --- | --- | --- | --- | --- | --- | --- | --- | --- |
| | $V$ | $\rho$ | $T$ | $e$ | | | $V$ | $\rho$ | $T$ | $e$ | |
| 1 | 1 | 1 | 1 | 1 | 717.88 | 33 | 5 | 1 | 5 | 2 | 869.88 |
| 2 | 1 | 2 | 2 | 2 | 1030.80 | 34 | 5 | 2 | 6 | 1 | 888.56 |
| 3 | 1 | 3 | 3 | 3 | 187.04 | 35 | 5 | 3 | 7 | 4 | 457.92 |
| 4 | 1 | 4 | 4 | 4 | 309.32 | 36 | 5 | 4 | 8 | 3 | 630.40 |
| 5 | 1 | 5 | 5 | 5 | 325.68 | 37 | 5 | 5 | 1 | 6 | 255.24 |
| 6 | 1 | 6 | 6 | 6 | 371.40 | 38 | 5 | 6 | 2 | 5 | 441.52 |
| 7 | 1 | 7 | 7 | 7 | 436.44 | 39 | 5 | 7 | 3 | 8 | 469.88 |
| 8 | 1 | 8 | 8 | 8 | 791.60 | 40 | 5 | 8 | 4 | 7 | 519.20 |
| 9 | 2 | 1 | 2 | 3 | 696.52 | 41 | 6 | 1 | 6 | 4 | 820.04 |
| 10 | 2 | 2 | 1 | 4 | 1075.16 | 42 | 6 | 2 | 5 | 3 | 799.80 |
| 11 | 2 | 3 | 4 | 1 | 326.76 | 43 | 6 | 3 | 8 | 2 | 363.64 |
| 12 | 2 | 4 | 3 | 2 | 295.84 | 44 | 6 | 4 | 7 | 1 | 468.60 |

**Table 2.** *Cont.*

| Test Number | Factor Level | | | Empty Column | IMSP Value (kPa) | Test Number | Factor Level | | | Empty Column | IMSP Value (kPa) |
|---|---|---|---|---|---|---|---|---|---|---|---|
| | V | ρ | T | e | | | V | ρ | T | e | |
| 13 | 2 | 5 | 6 | 7 | 405.80 | 45 | 6 | 5 | 2 | 8 | 415.92 |
| 14 | 2 | 6 | 5 | 8 | 574.20 | 46 | 6 | 6 | 1 | 7 | 525.88 |
| 15 | 2 | 7 | 8 | 5 | 651.32 | 47 | 6 | 7 | 4 | 6 | 619.52 |
| 16 | 2 | 8 | 7 | 6 | 799.40 | 48 | 6 | 8 | 3 | 5 | 562.60 |
| 17 | 3 | 1 | 3 | 6 | 938.96 | 49 | 7 | 1 | 7 | 5 | 891.88 |
| 18 | 3 | 2 | 4 | 5 | 1000.28 | 50 | 7 | 2 | 8 | 6 | 1150.84 |
| 19 | 3 | 3 | 1 | 8 | 231.96 | 51 | 7 | 3 | 5 | 7 | 365.88 |
| 20 | 3 | 4 | 2 | 7 | 376.12 | 52 | 7 | 4 | 6 | 8 | 420.48 |
| 21 | 3 | 5 | 7 | 2 | 463.60 | 53 | 7 | 5 | 3 | 1 | 634.56 |
| 22 | 3 | 6 | 8 | 1 | 466.40 | 54 | 7 | 6 | 4 | 2 | 567.28 |
| 23 | 3 | 7 | 5 | 4 | 588.44 | 55 | 7 | 7 | 1 | 3 | 402.20 |
| 24 | 3 | 8 | 6 | 3 | 667.64 | 56 | 7 | 8 | 2 | 4 | 466.52 |
| 25 | 4 | 1 | 4 | 8 | 426.20 | 57 | 8 | 1 | 8 | 7 | 909.32 |
| 26 | 4 | 2 | 3 | 7 | 925.00 | 58 | 8 | 2 | 7 | 8 | 1116.48 |
| 27 | 4 | 3 | 2 | 6 | 311.40 | 59 | 8 | 3 | 6 | 5 | 390.84 |
| 28 | 4 | 4 | 1 | 5 | 388.48 | 60 | 8 | 4 | 5 | 6 | 384.04 |
| 29 | 4 | 5 | 8 | 4 | 379.04 | 61 | 8 | 5 | 4 | 3 | 646.20 |
| 30 | 4 | 6 | 7 | 3 | 344.92 | 62 | 8 | 6 | 3 | 4 | 397.24 |
| 31 | 4 | 7 | 6 | 2 | 592.48 | 63 | 8 | 7 | 2 | 1 | 572.80 |
| 32 | 4 | 8 | 5 | 1 | 613.24 | 64 | 8 | 8 | 1 | 2 | 612.36 |

In orthogonal experiments analysis, an empty column is often used as the error column, which is the sum of the effects on hardness by other factors, except those already proposed (i.e., density, temperature and punch head velocity). Error analysis can be used to describe the significance of factors in the variance analysis of orthogonal experiments [18].

*3.3. Range Analysis Results and Discussions*

The orthogonal table states that range analysis allows one to assess the sensitivity of each factor to the IMSP Value [19].

In reference to the general method of orthogonal experimental processing, the IMSP Value of $i$th-factor and $j$th-level number is specified as $F_i^j$. Furthermore, the total average of all data, $\overline{F}(X)$, can be calculated:

$$\overline{F}(X) = \frac{1}{64} \sum_{i=1}^{3} \sum_{j=1}^{8} F_i^j = 574.17 \text{kPa}, \tag{1}$$

Take the average of all the IMSP Value that appear at $j$th-level of $i$th-factor in the experimental design, which is indicated as the average level value of $i$th-factor at $j$th-level, denoted as $\overline{F}_i^j$. The range of $i$th-factor's effect on the hardness of compacted snow ($\Delta F_i$) is defined by the difference between the maximum and minimum values among $\overline{F}_i^j$, and $j$ is from 1 to 8. Table 3 represents the range analysis of the experimental results.

**Table 3.** Mean and range analysis (kPa) of orthogonal test.

| Factors | $\overline{F}_i^1$ | $\overline{F}_i^2$ | $\overline{F}_i^3$ | $\overline{F}_i^4$ | $\overline{F}_i^5$ | $\overline{F}_i^6$ | $\overline{F}_i^7$ | $\overline{F}_i^8$ | $\Delta F_i$ |
|---|---|---|---|---|---|---|---|---|---|
| V | 521.27 | 603.13 | 591.68 | 497.60 | 566.57 | 572.01 | 612.46 | 628.66 | 131.07 |
| ρ | 783.84 | 998.36 | 329.43 | 409.16 | 440.75 | 461.11 | 541.64 | 629.07 | 668.94 |
| T | 526.14 | 538.95 | 551.39 | 551.85 | 565.14 | 569.66 | 622.41 | 667.83 | 141.69 |

The results are directly compared with the previously reported findings on snow hardness. Zhang conducted measurements of the compacted snow in Harbin utilizing the IMSP [15]. In Figure 4.5 (a) of reference [15], the hardness value of hexagonal dendritic snow crystals is measured to be approximately 490 kPa at the density of 400 kg/m$^3$. This is similar to the $\overline{F}_\rho^7$ presented in Table 3 of this study (i.e., the average hardness of 402 kg/m$^3$ of snow is 541.64 kPa within the experimental range). This observation suggests that the orthogonal test analysis maintains a certain degree of reliability.

The range of sample density, $\Delta F_\rho$, is 668.94 kPa, which is significantly greater than the ranges of sample temperature (141.69 kPa) and cone penetration speed (131.07 kPa). The range analysis of various factors reveals that, within the factor range of this experiment, the impact of compacted snow density on hardness is by far the greatest, surpassing the impact of temperature and penetration speed.

### 3.4. Variance Analysis

The range analysis method does not account for the effects of data variations carried on by the testing circumstances, nor does it provide a standard for evaluating the significance of each factor's influence [20]. In contrast to the range analysis method, the variance analysis method appropriately corrects this weakness [21,22].

Using the quantitative F-test results, it can be determined if the effects of particular factor is significant. The following are the fundamental steps of variance analysis [23].

For a particular factor $i$, let $SS_i$ be

$$SS_i = \frac{1}{8}\sum_{j=1}^{8}\left(8\overline{F}_i^j\right)^2 - \frac{1}{64}\left[64\overline{F}(X)\right]^2$$

$$= 8\sum_{j=1}^{8}\left(\overline{F}_i^j\right)^2 - 64\left[\overline{F}(X)\right]^2, \ (i = V, \rho, T) \tag{2}$$

$SS_i$ signifies the effect of the $i$th-factor on the experimental results, which is also known as the sum of squares of $i$th-factor's deviations.

The expression for the variance of the $i$th-factor (i.e., $V_i$) is as follows:

$$V_i = \frac{SS_i}{df_i}, \tag{3}$$

The degree of freedom of the $i$th-factor, which is 7 in this study, is denoted by $df_i$.

To increase the reliability of the F-test, $V_i$ should be evaluated against $V_e$, calculating $FTEST_i$ (empty column in the orthogonal table serves to show the experimental error variance $V_e$ for this study). The $FTEST$ for the $i$th-factor is as follows:

$$FTEST_i = \frac{V_i}{V_e}, \tag{4}$$

Table 4 lists the analysis results for variance.

**Table 4.** Analysis results for variance calculated from every factor and error term.

| Sources | $SS_i$ | $df_i$ | $V_i$ | $FTEST_i$ |
|---------|--------|--------|-------|-----------|
| $V$ | 114,441.90 | 7 | 16,348.84 | 4.44 |
| $\rho$ | 2,765,486.06 | 7 | 395,069.44 | 107.35 |
| $T$ | 126,117.76 | 7 | 18,016.82 | 4.90 |
| $e$ | 25,760.88 | 7 | 3680.13 | —— |

Figure 5 shows the result of $FTEST_i$ with $F\alpha(7, 7)$ for comparison.

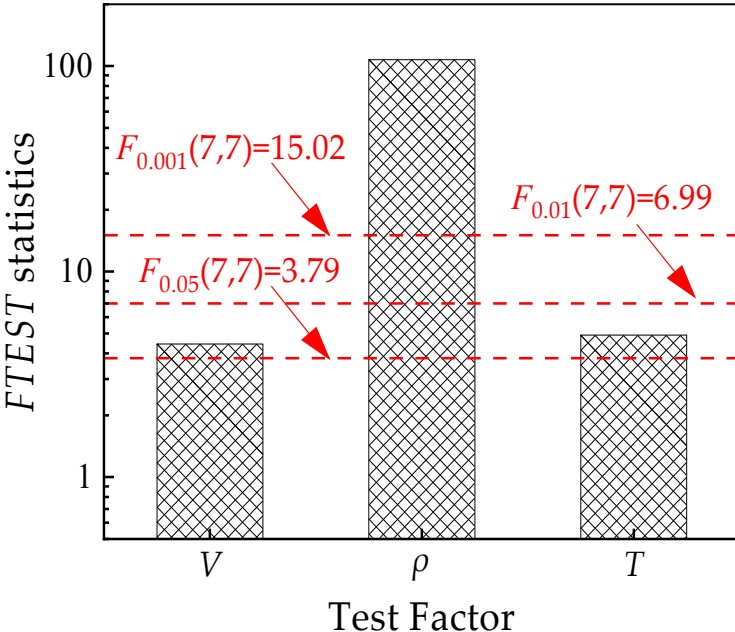

**Figure 5.** *FTEST* of various factors and *Fα*(7, 7) comparison results. A higher *FTEST* than *Fα*(7, 7) indicates that this factor is significant, at least at an inspection level of $\alpha$.

It can be observed that the penetration velocity and sample temperature *FTEST*s are both greater than $F_{0.05}(7, 7)$, so it can be concluded that the penetration velocity and sample temperature are significant at a 0.05 inspection level. And the *FTEST* of the sample density is greater than $F_{0.001}(7, 7)$, so it can be concluded that, at an inspection level of 0.001, the sample density is statistically significant.

The results of the analysis of variance indicate that:

1. Sample density has the most significant influence on the hardness of the sample;
2. The penetration velocity and sample temperature are significant at an inspection level of 0.05 and cannot be ignored.

## 4. Support Vector Regression (SVR) Algorithm for Model Development

Obtaining a continuous fitting function between each factor and the output variable, the IMSP Value, will make it more straightforward to study the properties that affect snow hardness within the experimental range. This can be an important addition to orthogonal experiment analysis. Numerous academic works have employed the orthogonal experiment-SVR to investigate different issues, obtaining favorable achievements [24,25]. To accomplish continuous fitting, this article employs the Support Vector Regression (SVR), which has multiple advantages and is widely utilized in regression analysis [26,27].

### 4.1. The Selection of Kernel Function

The selection of kernel function types is the most widely studied problem within the field of SVR model research. This study employs a Gaussian Radial Basis Function (RBF) kernel. The Gaussian RBF kernel can suit multiple functions more precisely than Linear SVR, Sigmoid SVR, and Polynomial SVR [28].

Equation (5) represents the fitting function of this SVR model, which combines the Gaussian RBF kernel function and the support coefficient.

$$F = \sum_{i=1}^{n} k_i \cdot G(X, X\prime), \tag{5}$$

In this experiment, *X* is the support vector for the algorithm; $G(X, X')$ is the mapping of each support vector under the Gaussian RBF kernel; and $k_i$ is the support coefficient for each Gaussian RBF kernel.

Equation (6) expresses the kernel function of the Gaussian distribution.

$$G(X, X') = e^{-\frac{\|X-X'\|^2}{2\sigma^2}},$$ (6)

### 4.2. The Particle Swarm Optimization-Support Vector Regression (PSO-SVR) Parameter Optimization Architecture and Proceeding

In this research, the PSO method, which is often used in SVR, is applied to optimize the penalty factor *c* and the SVR model parameter *γ* [29–31].

In actual operation, parameter optimization is accomplished with the architecture depicted in Figure 6.

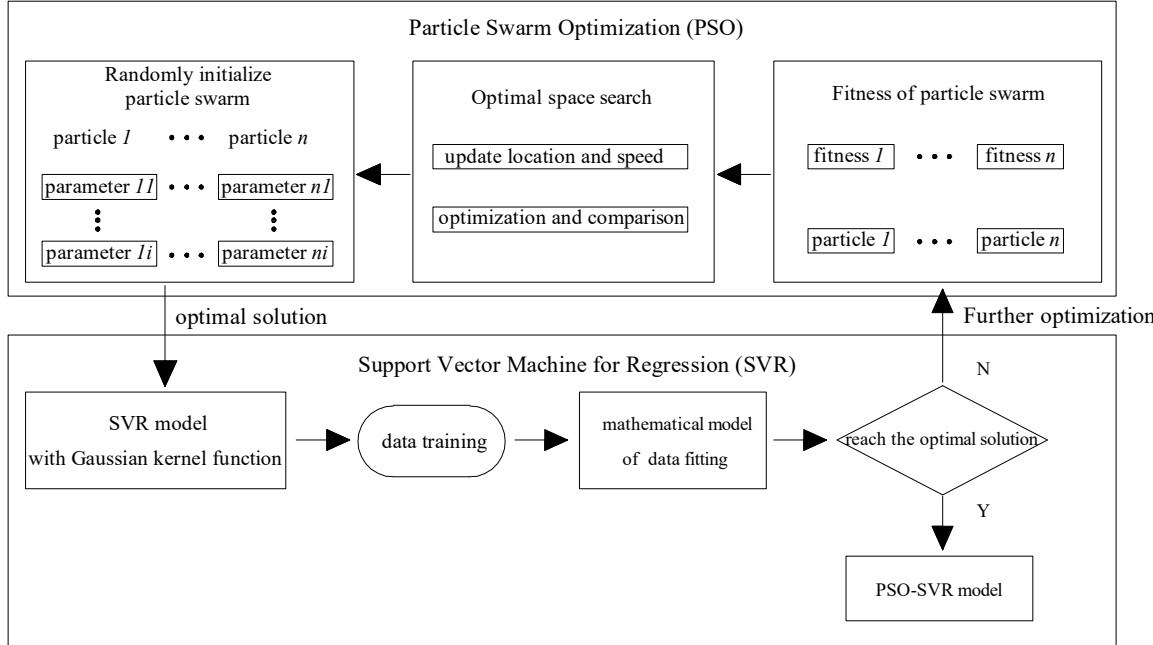

**Figure 6.** PSO-SVR architecture to optimize the penalty factor *c* and the SVR model parameter *γ*.

The optimization steps of PSO-SVR are shown as follows:

1. Initialize all particle positions and speeds at random;
2. Based on the fitness function of the snow hardness estimate issue, the fitness value of each particle is determined;
3. The fitness value of each particle is compared with both *Pbest* (i.e., the best position visited by this particle so far) and *Gbest* (i.e., the best position found by all the particles so far). In the event that the fitness value surpasses the current value of *Pbest*, it is necessary to update *Pbest* with the new fitness value. In the event that the fitness value surpasses the current value of *Gbest*, the *Gbest* should be updated with the new fitness value;
4. In order to expand the space for particles and prevent convergence to local optima, it is necessary to reset the penalty factor *c* of the SVR on each update;
5. Update the position and speed of every individual particle until the predetermined maximum number of iterations has been attained. Subsequently, output the optimal parameters. Alternatively, go back to Step 2.

### 4.3. The Result of PSO-SVR Algorithm

With 64 orthogonal experiment results (i.e., Table 2), the PSO-SVR algorithm is applied to the experiment to fit the continuous function of the three factors (i.e., density, temperature and punch head velocity) and the IMSP Value. The support vector and coefficients must be calculated first. Each input factor must be adjusted prior to training $\omega = (X, t)$ in order to set the iteration range of the Gaussian RBF kernel function parameter and penalty factor $c$. The output value $G$ is normalized based on Equation (7), with the normalization interval $[y_{\min}, y_{\max}]$ set to $[-1, 1]$. Equation (8) represents the function expression fitted by the SVR, where $\omega$ is the support vector, $n$ is the number of support vectors, $k_i$ is the coefficient corresponding to the $i$-th support vector Gaussian RBF kernel, and $b$ is the constant term of the function. Through reverse normalization, the ultimate predicted value $G$ is obtained.

$$\begin{cases} \omega' = \frac{y_{\max} - y_{\min}}{\omega_{\max} - \omega_{\min}}(\omega - \omega_{\min}) + y_{\min} \\ G' = \frac{y_{\max} - y_{\min}}{G_{\max} - G_{\min}}(G - G_{\min}) + y_{\min} \end{cases}, \tag{7}$$

$$G' = \sum_{i=1}^{n} k_i \exp(-\gamma \|\omega' - \omega'_*\|^2) + b, \tag{8}$$

58 support vectors are identified among the final 64 training sample points. Table 2 can be utilized as training samples to determine Table A1 in Appendix A. Table A1 provides a listing of the normalized support vectors and coefficient terms, with a constant term $b$ of $-0.224$.

The normalized support vector, corresponding coefficient term, and constant term $b$ are introduced into Equation (8) to determine the continuous function relationship between the three experimental factors and the output variable, the IMSP Value (i.e., the SVR model).

Figure 7 depicts a comparison between the predict values obtained from algorithm training and the true experimental values.

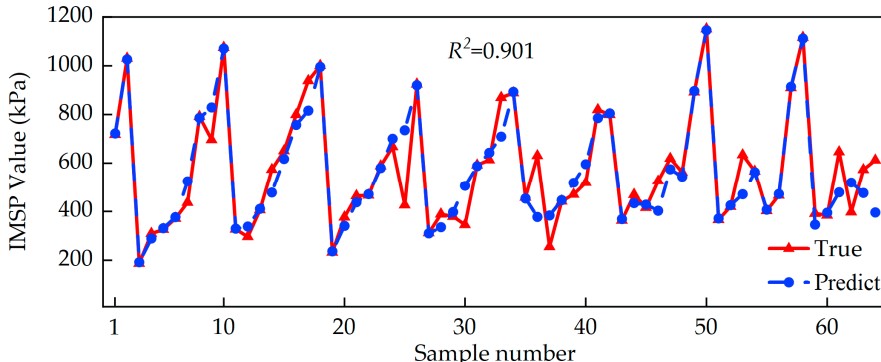

**Figure 7.** Comparison of true value and prediction value. Red line is the true IMSP Value, while blue line is the prediction value by SVR model. This comparison demonstrates that the trained SVR model is extremely close to the actual values, showing the accuracy performance of this SVR model.

The SVR model can get a minimal deviation between the actual value and the predict value, and this model has good nonlinear fitting ability. After seeking for parameters by PSO optimization, the SVR model parameter $\gamma$ in this study is 0.497, and the penalty factor $c$ is 8.594. The fitness parameter is 0.025, and the $R^2$ value is 0.901.

At a particular penetration velocity, using temperature and density as horizontal coordinates and the IMSP Value as the vertical coordinates, the coordinate system of SVR model can be drawn as Figure 8.

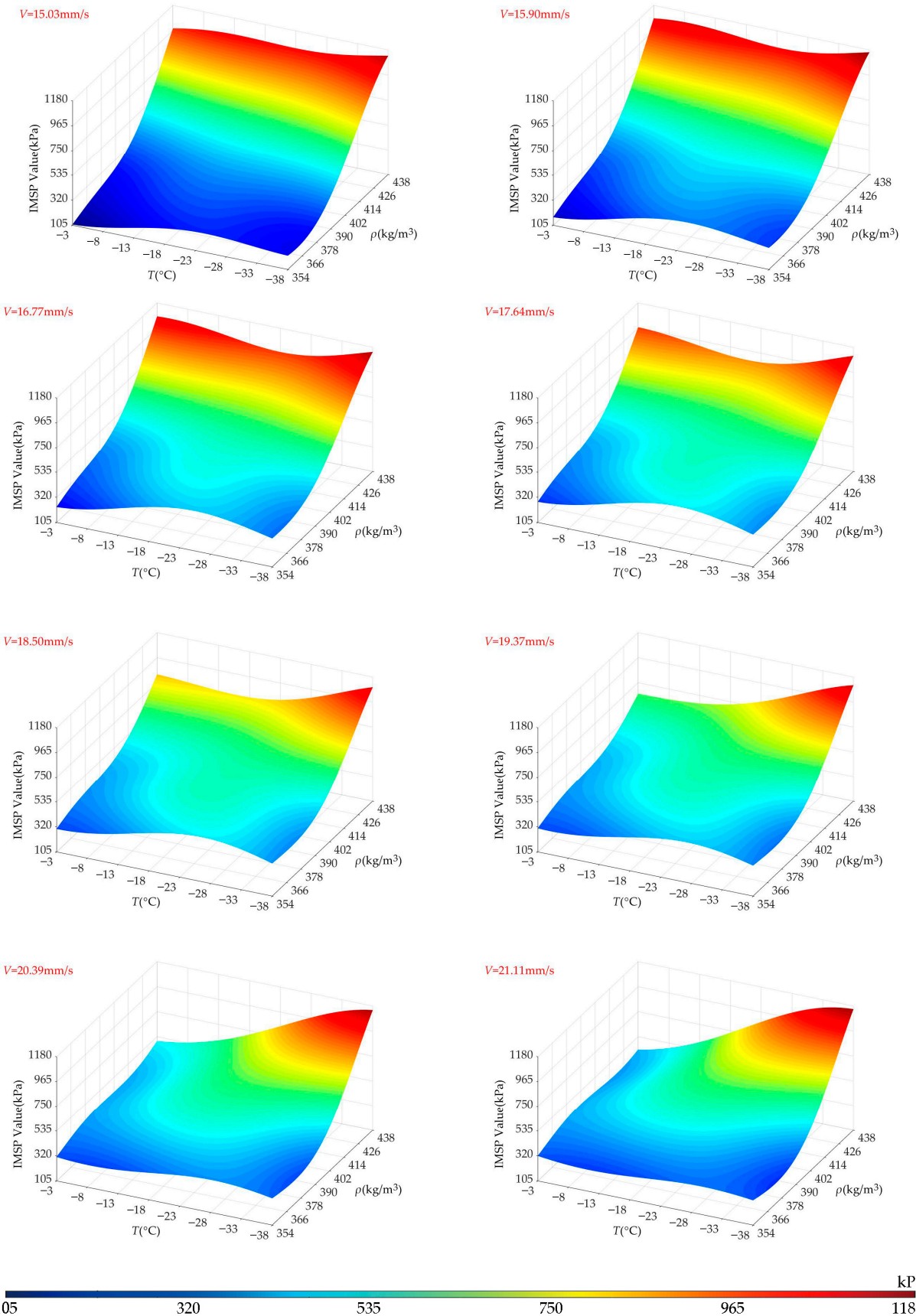

**Figure 8.** SVR model of relationship between the IMSP Value, temperature, and density with different penetration velocities.

According to Figure 8, Our most intriguing finding is the relationship between compacted snow temperature and hardness. On the one hand, when the density is low and the penetration speed is constant, the hardness shows a trend to rise first and then drop as the temperature falls. On the other hand, when the density is high, the hardness shows a trend to decline first and then increase as the temperature decreases.

Nevertheless, according to Figure 8, the change of the hardness brought about by temperature is significantly less than that by density. The relationship between compacted snow density and hardness is quite substantial. This result is similar to that reported by orthogonal experiment. In terms of overall trend, except for individual special cases, there is a significant positive correlation between the hardness and density of compacted snow. Both the conventional and this investigation illustrate similar performance [32]. A notable exception is that when the density of compacted snow is low, there is no significant positive correlation between snow density and hardness at specific temperatures and punch head speeds. It is suspected that this may be due to the influence of other factors on snow hardness. Under conditions which the compacted snow density is higher than 400 kg/m$^3$, the density is the most dominant factor that affects compacted snow hardness. Furthermore, it is necessary to guarantee that the snow density is sufficiently high, otherwise it is difficult to obtain compacted snow structures that meet the hardness requirements.

It is worth noting that if the penetration speed is high and the temperature is near the melting point, the hardness of the snow sample is very low, even if its density is high. This study suggests that compacted snow structures should avoid loading, particularly rapid loading, as much as possible when approaching the snow's melting point.

## 5. Conclusions

In order to investigate which one among density, temperature and punch head velocity is more sensitive and significant for compacted snow hardness measured by the IMSP, this study employs orthogonal experiments. The PSO-SVR algorithm was adopted to obtain further in-depth between the IMSP Value and these three factors, especially to explore the special case of a sharp decrease of snow hardness. For compacted snow structures represented by ski slopes, this investigation can provide some useful information for construction and usage, especially for structural safety. Specifically, the following are drawn as conclusions:

1.  This study uses the orthogonal experiment method, and orthogonal experiments are conducted on 64 different samples of compacted snow. The results of the range analysis indicate that the density of compacted snow samples has the most sensitive impact on hardness. Temperature and penetration velocity have far less sensitive effect on hardness than density.

2.  According to the variance analysis of the orthogonal experiment, the effect of density on hardness within the range of this experiment is the most significant. Comparatively, the effects of temperature and penetration velocity are limited to the inspection level of 0.05, which cannot be ignored completely. The significance of density is clearly supported by the current findings.

3.  Employing PSO-SVR analysis, we obtain the continuous function relationship between the IMSP Value and the three factors (i.e., density, temperature, penetration velocity), within the experimental range. This study not only confirms the positive correlation between density and hardness [32], but also discovers the relationship between temperature and hardness. By carefully examining the model, it is found that if the density of compacted snow is low, the hardness of the snow also tends to remain low. In this case, there is no significant positive relationship between snow density and hardness at specific temperatures and punch head speeds. Therefore, when constructing compacted snow roads, it is necessary to avoid the occurrence of weak areas of low-density snow. One of the more significant findings to emerge from this study is that when the temperature approaches the melting point of snow, even with high density, the hardness remains low, which is especially apparent at high

penetration velocities. This indicates that the hardness of snow suffers a fundamental decrease at high temperatures. In actual compacted snow road projects, when the temperature approaches 0 °C, even if the density of the compacted snow road is high, maintenance of the piste is still required. In this case, skiing is never permitted because of this sharp decrease in hardness, which could cause serious safety issues.

4. The IMSP employs an electric motor to precisely control the penetration speed, and utilizes the sensors to precisely measure the end snow resistance. This device can be used as a penetrating instrument on pistes and lays the groundwork for future research.

**Author Contributions:** Conceptualization, Z.L.; methodology, Z.L. and Q.Z.; investigation, Q.Z., S.H. and J.W.; data curation., S.H. and J.W.; formal analysis, Q.Z. and H.Y.; visualization, Q.Z. and H.Y.; writing—original draft preparation, Q.Z.; writing—review and editing, Q.Z., P.L., Q.W. and H.Y.; project administration, Z.L., P.L. and Q.W. All authors have read and agreed to the published version of the manuscript.

**Funding:** The Major Scientific and Technological Projects of the Ministry of Water Resources of China (SKS-2022017).

**Institutional Review Board Statement:** Not applicable.

**Informed Consent Statement:** Not applicable.

**Data Availability Statement:** The data are available in the case that it is required.

**Conflicts of Interest:** The authors declare no conflict of interest.

**Appendix A**

**Table A1.** Coefficient terms and normalized support vectors.

| Number | Coefficient Term | Super Vector | | | Number | Coefficient Term | Super Vector | | |
|---|---|---|---|---|---|---|---|---|---|
| | | $V$ | $\rho$ | $T$ | | | $V$ | $\rho$ | $T$ |
| 1 | −2.773 | −1.000 | 0.714 | 1.000 | 30 | −8.594 | −0.141 | 0.429 | −0.143 |
| 2 | 0.615 | −1.000 | 1.000 | 0.714 | 31 | 8.594 | 0.141 | 0.714 | −0.143 |
| 3 | −3.694 | −1.000 | −1.000 | 0.429 | 32 | −5.231 | 0.141 | 1.000 | −0.429 |
| 4 | 8.594 | −1.000 | −0.714 | 0.143 | 33 | 0.681 | 0.141 | −1.000 | −0.714 |
| 5 | −0.921 | −1.000 | −0.429 | −0.143 | 34 | 8.594 | 0.141 | −0.714 | −1.000 |
| 6 | −4.251 | −1.000 | −0.143 | −0.429 | 35 | −8.594 | 0.141 | −0.429 | 1.000 |
| 7 | −8.594 | −1.000 | 0.143 | −0.714 | 36 | −0.577 | 0.141 | −0.143 | 0.714 |
| 8 | 0.134 | −1.000 | 0.429 | −1.000 | 37 | −8.594 | 0.141 | 0.143 | 0.429 |
| 9 | −8.594 | −0.714 | 0.714 | 0.714 | 38 | −8.594 | 0.141 | 0.429 | 0.143 |
| 10 | 5.390 | −0.714 | 1.000 | 1.000 | 39 | 8.594 | 0.428 | 0.714 | −0.429 |
| 11 | −8.594 | −0.714 | −0.714 | 0.429 | 40 | −8.165 | 0.428 | 1.000 | −0.143 |
| 12 | −7.315 | −0.714 | −0.429 | −0.429 | 41 | −8.147 | 0.428 | −1.000 | −1.000 |
| 13 | 8.594 | −0.714 | −0.143 | −0.143 | 42 | 8.594 | 0.428 | −0.714 | −0.714 |
| 14 | 8.594 | −0.714 | 0.143 | −1.000 | 43 | −8.594 | 0.428 | −0.429 | 0.714 |
| 15 | 8.594 | −0.714 | 0.429 | −0.714 | 44 | 8.594 | 0.428 | −0.143 | 1.000 |
| 16 | 8.594 | −0.428 | 0.714 | 0.429 | 45 | 8.594 | 0.428 | 0.143 | 0.143 |
| 17 | 4.388 | −0.428 | 1.000 | 0.143 | 46 | 8.594 | 0.428 | 0.429 | 0.429 |
| 18 | −4.978 | −0.428 | −1.000 | 1.000 | 47 | −5.572 | 0.763 | 0.714 | −0.714 |
| 19 | 8.594 | −0.428 | −0.714 | 0.714 | 48 | 6.969 | 0.763 | 1.000 | −1.000 |
| 20 | 8.594 | −0.428 | −0.429 | −0.714 | 49 | −3.816 | 0.763 | −1.000 | −0.143 |

**Table A1.** *Cont.*

| Number | Coefficient Term | Super Vector | | | Number | Coefficient Term | Super Vector | | |
|---|---|---|---|---|---|---|---|---|---|
| | | V | $\rho$ | T | | | V | $\rho$ | T |
| 21 | −2.825 | −0.428 | −0.143 | −1.000 | 50 | −4.590 | 0.763 | −0.714 | −0.429 |
| 22 | 8.594 | −0.428 | 0.143 | −0.143 | 51 | 8.594 | 0.763 | −0.429 | 0.429 |
| 23 | −8.594 | −0.428 | 0.429 | −0.429 | 52 | 1.699 | 0.763 | −0.143 | 0.143 |
| 24 | −8.594 | −0.141 | 0.714 | 0.143 | 53 | −1.500 | 0.763 | 0.143 | 1.000 |
| 25 | 3.708 | −0.141 | 1.000 | 0.429 | 54 | −5.748 | 0.763 | 0.429 | 0.714 |
| 26 | 8.594 | −0.141 | −0.714 | 1.000 | 55 | −6.869 | 1.000 | 0.714 | −1.000 |
| 27 | −8.594 | −0.141 | −0.429 | −1.000 | 56 | 6.076 | 1.000 | 1.000 | −0.714 |
| 28 | −8.594 | −0.141 | −0.143 | −0.714 | 57 | 8.594 | 1.000 | −1.000 | −0.429 |
| 29 | 4.343 | −0.141 | 0.143 | −0.429 | 58 | −8.594 | 1.000 | −0.714 | −0.143 |

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
