# Peer review of "An Investigation of the Influence on Compacted Snow Hardness by Density, Temperature and Punch Head Velocity"

_water, doi:10.3390/w15162897_

Round 1

Reviewer 1 Report

This study investigated the relationship between compacted snow hardness and three factors (density, temperature, and penetration velocity) based on fixed-point control experiments. Overall, the article structure, language expression, and experimental design are relatively reasonable, the main concern is as follows:

    1. Section 4.2 only briefly introduces the PSO-SVR algorithm itself, and it is recommended to elaborate on the specific process in conjunction with the Snow Hardness estimation problem in this study. In addition, 64 samples were all used as training samples. The author should consider N-fold cross validation or dividing the samples into training and testing samples to evaluate the performance of the developed model.

2. The universality of experimental conclusions is a problem because this study is a control experiment under ideal conditions, while the compaction of snow roads under natural conditions is much more complex.

Minor comments:

1. Line 35: N.R.C., what is mean?

2. Line 37: remove “formulas for”. In fact, there are still many similar issues. It is recommended to read the entire text again and make revisions.

3. Line 45:old machine-made snow has lower strength than new machine-made snow. Should old/new machine-made snow be machine-made old/new snow?

4. Line 227: What does “SVR model parameter” specifically refer to? It should be clearly stated.

Most sentences have good language expression,individual sentences need to be revised.

Reviewer 2 Report

The manuscript addresses an analysis of the density, temperature, and speed of the punch head to analyze the variation in the hardness of compacted snow measured by penetrometers. The manuscript is very well-written, with statistical analyses properly presented.

I only suggest that the authors improve the discussion of their results by analyzing and comparing them with other studies related to the subject. This will demonstrate the unprecedented contribution of this research.

Reviewer 3 Report

1. In line 31, should it be the actual area of contact as generally it is considered for evaluating frictional properties of contact?

2. In line 35, it would be beneficial for the reader to use the full form of N.R.C. instead of the acronym the first time.

3. In line 46, I am assuming you mean compressive strength but it would be more helpful if mentioned outright. Also, what density range are you considering as the behavior in the fresh snow domain and compacted snow domain might not be similar?

4. Regarding the cone description in line 97: In literature studies, it has been found that the 30-degree cone angle (Russian Snow Penetrometer) was more beneficial than the 60-degree cone angle (Rammsonde) for compacted snow. Is there a reason for utilizing the 60-degree cone angle?

5. In line 110, it would be better if the reason for 28 days of metamorphosis is stated at that time instead of the succeeding paragraph.

6. Maybe it would be beneficial for the reader to know the concept of error level and how different test numbers have the same error level number before Table 2. Does it mean the error was the same in those instances?

7. Towards the end of section 4 or in section 5, it would be beneficial if the authors mention how the findings of this research could help in compacted snow applications like skiing, transportation, etc. Maybe it could be of help to institutional/regulatory bodies to create safety guidelines/protocols.

1. There are no spelling errors but it would be beneficial for the readers to proofread the document for grammatical accuracy.

2. It would be beneficial for the reader if the grammar of the last paragraph of section 2 in the start of page 4 is improved as it is a crucial one describing the working of the device.
